# Prenatal Diagnosis of Jeune Syndrome Caused by Compound Heterozygous Variants in *DYNC2H1* Gene—Case Report with Rapid WES Procedure and Differential Diagnosis of Lethal Skeletal Dysplasias

**DOI:** 10.3390/genes13081339

**Published:** 2022-07-27

**Authors:** Agnieszka Stembalska, Małgorzata Rydzanicz, Magdalena Klaniewska, Lech Dudarewicz, Agnieszka Pollak, Mateusz Biela, Piotr Stawinski, Rafal Ploski, Robert Smigiel

**Affiliations:** 1Department of Genetics, Medical University, Marcinkowskiego 1, 50-368 Wroclaw, Poland; agnieszka.stembalska@umw.edu.pl; 2Department of Medical Genetics, Medical University of Warsaw, Adolfa Pawińskiego 3c, 02-106 Warsaw, Poland; mrydzanicz@wum.edu.pl (M.R.); apollak@wum.edu.pl (A.P.); piotr.stawinski@wum.edu.pl (P.S.); rploski@wp.pl (R.P.); 3Department of Family and Pediatric Nursing, Medical University, Bartla 5, 51-618 Wroclaw, Poland; magdazdzie@gmail.com (M.K.); mateuszbiela14@gmail.com (M.B.); 4Department of Genetics, Polish Mother’s Memorial Hospital-Research Institute, Rzgowska 281/289, 93-338 Lodz, Poland; lechdudarewicz@gmail.com

**Keywords:** asphyxiating thoracic dystrophy, Jeune syndrome, *DYNC2H1*, lethal skeletal dysplasia, life-limiting skeletal dysplasia, prenatal diagnosis

## Abstract

Skeletal dysplasias (SDs) are a large, heterogeneous group of mostly genetic disorders that affect the bones and cartilage, resulting in abnormal growth and development of skeletal structures. The high clinical and genetic diversity in SDs cause difficulties in prenatal diagnosis. To establish a correct prognosis and better management, it is very important to distinguish SDs with poor life-limiting prognosis or lethal SDs from other ones. Bad prognosis in foetuses is assessed on the basis of the size of the thorax, lung volumes, long bones’ length, bones’ echogenicity, bones’ angulation or presented fractures, and the concomitant presence of non-immune hydrops or visceral abnormalities. To confirm SD diagnosis and perform family genetic consultation, rapid molecular diagnostics are needed; therefore, the NGS method using a panel of genes corresponding to SD or whole-exome sequencing (WES) is commonly used. We report a case of a foetus showing long bones’ shortening and a narrow chest with short ribs, diagnosed prenatally with asphyxiating thoracic dystrophy, also known as Jeune syndrome (ATD; OMIM 208500), caused by compound heterozygous variants in the *DYNC2H1* gene, identified by prenatally performed rapid-WES analysis. The missense variants in the *DYNC2H1* gene were inherited from the mother (c.7289T>C; p.Ile2430Thr) and from the father (c.12716T>G; p.Leu4239Arg). The *DYNC2H1* gene is one of at least 17 ATD-associated genes. This disorder belongs to the ninth group of SD, ciliopathies with major skeletal involvement. An extremely narrow, bell-shaped chest, and abnormalities of the kidneys, liver, and retinas were observed in most cases of ATD. Next to lethal and severe forms, clinically mild forms have also been reported. A diagnosis of ATD is important to establish the prognosis and management for the patient, as well as the recurrence risk for the family.

## 1. Introduction

Skeletal dysplasias (SDs; osteochondrodysplasias) are a large, heterogeneous group of mostly genetic disorders with a prevalence of more than 1 in every 5000 newborns [1,2]. Due to the fact that some of the diseases in the SD group are fatal and also underdiagnosed, the incidence rate is probably higher in reality [1,3]. SDs affect the bones and cartilage, resulting in abnormal growth. Except for the lethal form, the phenotypic expression of SD typically continues to evolve throughout life [4].

Osteochondrodysplasias with many dysostoses are classified within the Nosology and Classification of Genetic Skeletal Disorders based on clinical, pathological, biochemical, and imaging methods (ultrasound/computed tomography/magnetic resonance), as well as molecular findings [3,5,6,7,8]. According to the Nosology of 2019, there are 461 different diseases classified into 42 groups [1,8,9]. It is very important to distinguish those SDs with poor life-limiting prognosis or that are lethal from other ones in order to establish a correct prognosis and better management [3,10]. The severity or lethality of SDs in the prenatal period are assessed on the basis of the size of the thorax, lung volumes, long bones’ length, bones’ echogenicity (i.e., mineralisation), bones’ angulation or presented fractures, and the concomitant presence of non-immune hydrops or visceral abnormalities [3,5,6]. The high clinical and genetic heterogeneity in SDs cause difficulties in prenatal diagnosis. However, recognising the correct form of SDs is important for determining prognosis, as well as for better management and family counselling [3,10,11]. Therefore, the NGS method using a panel of genes corresponding to SD or whole-exome sequencing (WES) are commonly used [6,12].

The asphyxiating thoracic dystrophy, also known as Jeune syndrome (ATD; OMIM 208500), belongs to the group of SDs named short-rib thoracic dysplasia (https://omim.org/graph/linear/PS208500, accessed on 1 July 2022).

Herein, we report a case of a foetus showing long bones’ shortening and a narrow chest with short ribs, diagnosed prenatally with asphyxiating thoracic dystrophy caused by compound heterozygous variants (NM_001080463.2:c.[7289T>C];[12716T>G]) in the *DYNC2H1* gene, identified by prenatally performed rapid-WES analysis.

## 2. Case Report

A 34-year-old woman was referred to genetic counselling at the beginning of the second trimester of her second pregnancy, conceived by IVF because of idiopathic infertility. The first pregnancy ended with spontaneous abortion at 8 weeks. The family history was negative.

Upon prenatal screening in the first trimester of the current pregnancy, the shortening of long bones was observed in another medical centre. The shortening of the long bones, narrowing of the chest with short ribs, bowing of the femur, excess soft tissues on the thighs and lower legs, suspicion of retrognathia, and ulnar deviation of the wrist were described on an ultrasound at 14 weeks of gestation plus 6 days (according to the date of oocyte retrieval) (Figure 1a,b).

On the basis of the ultrasound images, skeletal dysplasia with poor prognosis was suspected, with a probable diagnosis of type 1thanathophoric dysplasia. Therefore, amniocentesis and *FGFR3* gene sequencing or next-generation sequencing using a panel for skeletal dysplasias were proposed. After obtaining the patient’s informed consent, the amniocentesis under ultrasound guidance was performed in a typical way at 15 weeks of gestational age.

In the meantime, the next ultrasound scan was performed at the 17th week of gestation, with the aim of visualising the abnormal sulcation of the temporal lobes, which are pathognomonic of thanatophoric dysplasia (TD), as well as further detailing the phenotype. The observed sulcation pattern was normal. The ultrasound findings from the previous scan were generally confirmed; additionally, bilateral brachydactyly of the hands, mild polyhydramnios, moderate pericardial effusion, a small hydrothorax, and mild retrognathia without signs of the Pierre–Robin sequence were noted (Figure 2a,b). The shortening of the ribs was described as severe, with high probability of pulmonary hypoplasia. There was a protuberant abdomen and a relative excess of soft tissues of the lower and upper extremities attributable to the shortening of the limbs. The shortening of the long bones was assessed as severe (e.g., measured femur length vs. expected: 14.2 mm vs. 25 mm, respectively—three standard deviations below the median; fibula length: 9.2 mm—more than three standard deviations below the median) (Figure 2c,d). No other abnormalities were noticed.

After the rapid-WES result, the pregnancy was terminated by the intracervical administration of prostaglandins at 19 weeks of gestational age.

## 3. Genetic Testing

### Materials and Methods

Prenatal molecular diagnosis in the proband was performed using rapid-WES on the DNA isolated from amniotic cells with a Nextera Flex for Enrichment sample preparation kit (Illumina, San Diego, CA, USA) combined with TruSeq DNA Exome probes (Illumina). The enriched library was paired-end sequenced (2 × 100 bp) on a HiSeq 1500 (Illumina). Raw data analysis and variant prioritisation were performed according to the previously described pipeline [13]. Venous blood samples were collected from the proband’s parents for subsequent family studies. Variants considered as disease-causing were validated using DNA samples from the proband and proband’s parents by amplicon deep sequencing (ADS) performed using a Nextera XT Kit (Illumina) and paired-end sequenced (2 × 100 bp) on a HiSeq 1500 (Illumina).

## 4. Results

Rapid-WES is defined as a process of sample preparation and WES data analysis up to 48–72 h. The first WES result in our proband was available after 3 days. By applying WES analysis, in the proband, two heterozygous missense variants in the *DYNC2H1* gene were identified: (hg38, chr11:g.103188645-T>C, NM_001080463.2: c.7289T>C; p.Ile2430Thr, rs1221455921) and (hg38, chr11:g. 103468635-T>G, NM_001080463.2: c.12716T>G; p.Leu4239Arg, rs1945272232). In the proband, the presence of both variants was confirmed by ADS; thus, a compound heterozygote in *DYNC2H1* was observed. To determine if *DYNC2H1* compound heterozygous variants were in cis or in trans, a tri-based ADS family study was performed. It was found that the p.Ile2430Thr variant was inherited from the mother, while p.Leu4239Arg was inherited from the father (Figure 3), which is consistent with the in trans variants’ transmission in the autosomal recessive pattern of inheritance. The population frequency of c.7289T>C and c.12716T>G variants was 0 (gnomAD database v3.1.2; access 2 March 2022). According to ACMG classification, p.Ile2430Thr was classified as pathogenic, while p.Leu4239Arg was classified as a variant of uncertain significance [14].

## 5. Discussion

The vast majority of SDs occur without any known parental risk factors [3]. Therefore, in the case of significantly shortened limbs in any foetus, a concrete diagnosis of the type of SD is important. Lethality evaluation is significant for further management. It should be emphasised that the thorax circumference being below the fifth percentile for gestational age or a chest circumference-to-abdomen-circumference ratio below the fifth percentile predict pulmonary hypoplasia, which is a feature of life-limiting or lethal skeletal dysplasias [6,15].

Herein, we report a prenatally diagnosed case of ATD, also known as Jeune syndrome, due to compound heterozygous variants (p.Ile2430Thr and p.Leu4239Arg) in the *DYNC2H1* gene. The p.Leu4239Arg variant was previously reported to be associated with Jeune syndrome [12]. Meanwhile, p.Ile2430Thr is located in the AAA3 nucleotide-binding pocket (amino acids 2251–2505) of the *DYNC2H1* protein hexomeric ring-like ATP-hydrolysing motor domain (https://www.uniprot.org/uniprot/Q8NCM8, accessed on 2 March 2022) in close proximity with three known Jeune syndrome-associated variants, including p.Ser2423Tyr [16], described also as “pathogenic” in the ClinVar database (https://www.ncbi.nlm.nih.gov/clinvar, accessed on 2 March 2022), and p.Arg2426Cys [17] and p.Arg2426Leu [16,18]. Thus, taking into account the abovementioned facts, we consider both identified *DYNC2H1* p.Ile2430Thr and p.Leu4239Arg variants as pathogenic and causative.

*ATD* belongs to the group of SDs known as short-rib thoracic dysplasia (https://omim.org/graph/linear/PS208500, accessed on 1 July 2022). ATD affects an estimated 1 in 100,000 to 130,000 live births [19], and belongs to the ninth group of SD, ciliopathies with major skeletal involvement [9]. Ciliopathies are complex multisystem disorders of the cilia; therefore, all major organs, including the kidney, brain, eyes, lungs, and respiratory tract, as well as limbs, may be affected [20]. There are at least 17 ATD-associated genes (*DYNC2H1*, *DYNC2LI1*, *WDR34*, *TCTEX1D2*, *WDR60*, *WDR19*, *IFT140*, *TTC21B*, *IFT80*, *IFT172*, *IFT81*, *IFT52*, *TRAF3IP1*, *CFAP410*, *CEP120*, *KIAA0586*, and *KIAA0753*) encoding proteins involved in the formation or function of the cilia [9,21]. The *DYNC2H1* (dynein cytoplastic 2 heavy-chain 1) gene encodes the protein involved in ciliary intraflagellar transport (IFT)—the process required for ciliary assembly and maintenance (OMIM 603297). Cilia play a role in signalling pathways, such as the Sonic Hedgehog pathway, which is important for the growth, proliferation, and differentiation of cells, also those that give rise to cartilage and bone [12]. Moreover, variants in *DYNC2H1* were also described in types I-III short-rib polydactyly syndrome (SRPS) (Saldino–Noonan syndrome, Majewski syndrome, and Verma–Naumoff syndrome), which belong to the group of “ciliopathies with major skeletal involvement”, according to current nosology [22,23].

ATD is a rare autosomal, recessive skeletal dysplasia with additional involvement of multiple organs [7,24,25,26]. Skeletal abnormalities in ATD include short and wide ribs, a narrow, bell-shaped chest, and shortened long bones, leading to mild dwarfism [21]. In most cases of ATD, an extremely narrow, bell-shaped chest can restrict the growth and expansion of the lungs (rigid thoracic cavity). This severe form of Jeune syndrome may be lethal in the prenatal period or limit the survival of the neonate after birth [12,22,24]. In the presented case, the shortening of the long bones, narrowing of the chest with short ribs, bowing of the femur, excess of soft tissues on the thighs and lower legs, retrognathia, and ulnar deviation of the wrist were noted during ultrasound examination at 14 weeks of gestation. In prenatal diagnosis, short ribs and shortened limbs may be recognised first at 14 weeks gestation [24]. Other skeletal dysplasias with shortened ribs and a narrow thorax, which must be analysed in the prenatal differential diagnosis of ATD, are collected in Table 1 [27,28].

The degree of restriction of lung growth and expansion may vary in different cases of ATD, so the severity of respiratory distress, age of onset of symptoms, and prognosis may also vary [24,29,30]. Next to the lethal and severe forms, clinically mild forms have also been reported in the literature [12,29].

ATD is characterised also by other, less frequent skeletal changes, such as handle-bar-shaped clavicles, hypoplastic iliac wings, cone-shaped ends of the long bones in the arms and legs, short hands and feet, and spinal abnormalities with the most common proximal cervical stenosis [24,29]. A non-constant feature of ATD is polydactyly. Occasionally, postaxial hexadactyly may be observed [12,15].

In some cases, ATD may be associated with abnormalities of the kidneys (progressive renal disease), liver (hepatic fibrosis), retinas (retinal degeneration), and other tissues (pancreatic fibrosis, Hirschsprung disease, multiple gingival frenula, and mild hydrocephaly) [15,23]. Schmidts et al. described 29 patients with Jeune syndrome and biallelic variants in the *DYNC2H1* gene, and they noted low incidence of renal or retinal disease in these patients [12]. The authors suggested that lesions in these organs in Jeune syndrome due to variants in the *IF140* gene were more frequently described [12,31]. Lethality amongst JATD (Jeune Asphyxiating Thoracic Dystrophy) patients with dynein gene mutations, most frequently *DYNC2h1* variants, is highest due to the severe rib shortening not observed with *IFT* gene pathogenic variants. *DYNC2H1* JATD patients rarely exhibit the extraskeletal features found with *IFT* gene variants; however, they are not lethal.

They also suggest that the observed higher survival rate of patients with *DYNC2H1* gene variants compared with those carrying variants in other ATD-related genes may be due to the lower frequency of extra-skeletal features [12].

A diagnosis of ATD is important, not only because of the prognosis and management for the patient, but also because of the recurrence risk for the family. In autosomal recessive disorders, such as ATD, there is 25% risk of another child having the same disorder; therefore, prenatal or preimplantation diagnosis in the next pregnancy of the same parents could be offered [26]. Moreover, it should be emphasised that the prenatal use of NGS provides the possibility of quick diagnosis and, thus, quick implementation of appropriate management.

## Figures and Tables

**Figure 1 genes-13-01339-f001:**
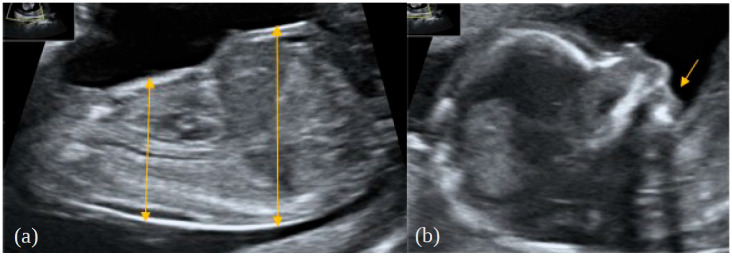
Foetal findings in the first trimester of the presented foetus; (**a**) narrow chest, protuberant abdomen; (**b**) retrognathia.

**Figure 2 genes-13-01339-f002:**
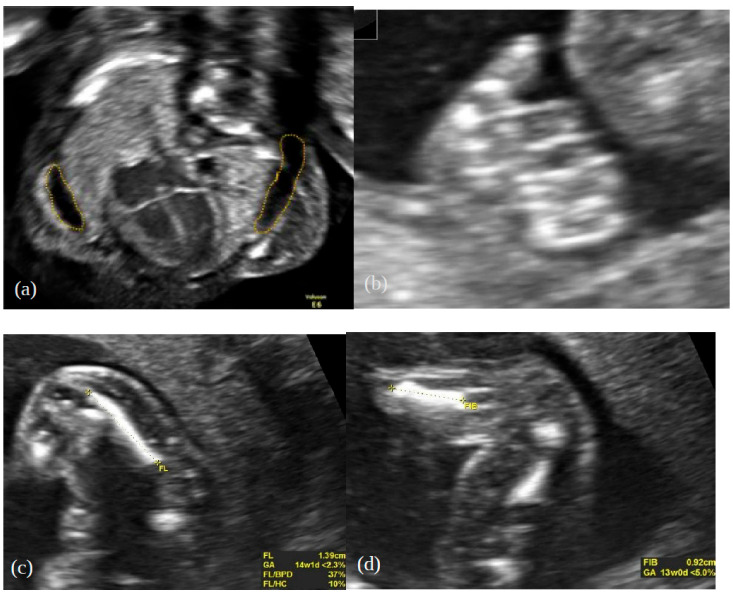
Prenatal features at 17 weeks of gestation: (**a**) hydrothorax, (**b**) brachydactyly, (**c**) shortened femur and (**d**) shortened fibula.

**Figure 3 genes-13-01339-f003:**
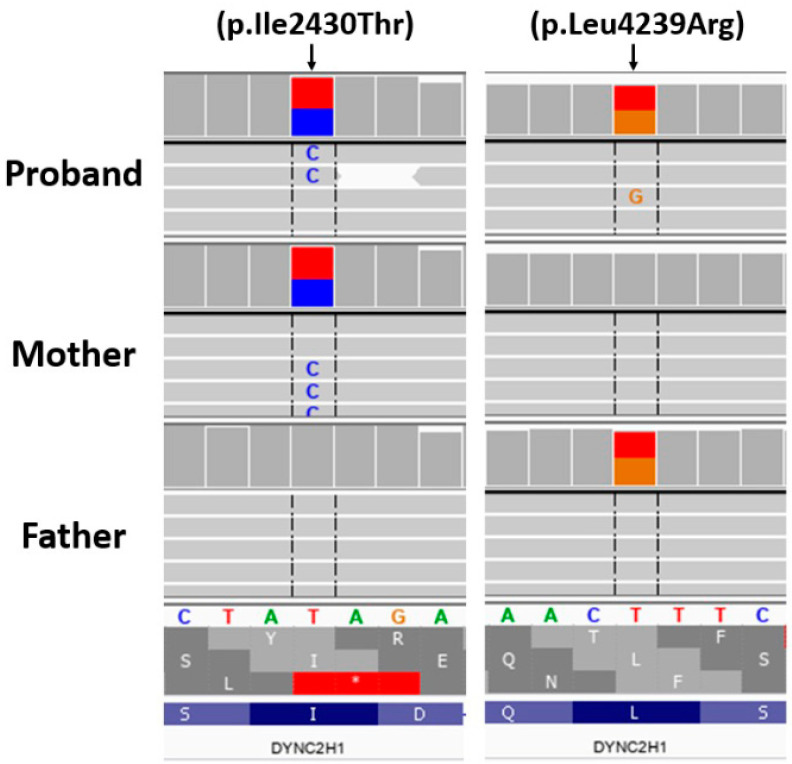
Variants of *DYNC2H1* confirmed by ADS in the proband and parents.

**Table 1 genes-13-01339-t001:** Differential diagnoses of asphyxiating thoracic dystrophy (ATD; OMIM 208500) with other skeletal dysplasias with shortened limbs, short ribs, and a narrow chest in the prenatal period.

Skeletal Dysplasia	Clinical Description	Inheritance
Short rib polydactyly ciliopathy syndromes	Constricted thoracic cage, short ribs, short tubular bones, trident appearance of the acetabular roof, polydactyly, and microglossiaSoldino–Noonan and Verma–Naumoff types: marked metaphyseal irregularities; Majewski type: very short, oval tibiae Non-skeletal manifestation: cleft palate, cleft lip, cystic kidneys, genital ambiguity, and foetal hydrops	AR
Ellis van Creveld	Short ribs, orodental, cardiac defects, fusion between the hamate and capitate, peculiar deformity of the proximal tibia, short middle phalanges, and postaxial interdigital polydactyly	AR
Campomelic dysplasia	Severe limb shortening, tibial bowing, small chest, flat, short vertebrae, hypoplastic scapulae, small iliac wings, and sex reversal	AD
Thanatophoric dysplasia type 1	Severe long bone rhizomelic shortening and sometimes bowing, small chest, frontal forehead prominence, and polyhydramnios Type 1—short, bent femurs like a “French telephone receiver” Type 2—cloverleaf skull	AD
Achondroplasia (homozygous)	Limb rhizomelic shortness	AD
Achondrogenesis	Short ribs, extremely short limbs, flat face, hydrops, decreased ossification of skull and vertebral bodies, and ossified pubic bones	AD
Atelosteogenesis	Severe shortening of limbs, hypoplasia of the humeri, femurs, thoracic spine, dislocated elbows, hips, knee, and possible lack of ossification of single hand bones	AD

## Data Availability

The data presented in this study are available on request from the corresponding author.

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
