# Peer review of "Prenatal Diagnosis of Jeune Syndrome Caused by Compound Heterozygous Variants in *DYNC2H1* Gene—Case Report with Rapid WES Procedure and Differential Diagnosis of Lethal Skeletal Dysplasias"

_genes, 2022, doi:10.3390/genes13081339_

Round 1

Reviewer 1 Report

The efforts taken to present the genetic report of this patient is appreciated. Rapid WES are the way forward; and the most crucial in clinical decision making.

Some minor suggestions:

please present the full form of all abbreviations when cited first (e.g. AS in introduction; ADS in the genetic methods used)

Line 69 has a grammatical error

The discussion can be cut down; and made to revolve only around the novelty of the finding and its rarity

Otherwise, the text is largely a review of literature on the genetics and clinicopathology of ATDs.

Author Response

Thank you for your review. 
We have made the suggested improvements to the manuscript. 

Reviewer 2 Report

In the present manuscript, it suggests that the in trans compound heterozygous variants of the DYNC2H1 gene, the responsible gene of the asphyxiating thoracic dystrophy (ATD, Jeune syndrome), may cause the ATD disease. The authors found two heterozygous variants in the prenatal diagnosis using rapid whole-exosome sequencing (WES) and confirmed that the variants are in trans compound heterozygous variants in trio amplicon deep sequencing (ADS). I recommend that this paper be accepted after minor revision. 

1. The authors used the different disease name, ATD and Jeune syndrome and mentioned ATD, also known as Jeune syndrome. If it is not needed to distinguish, it is better to use either one name.

2. The rapid WES has identified the compound heterozygous variants, however it has not yet been determined whether they are in trans or in cis compound heterozygous variants. It is possible that in cis compound heterozygous variants are not pathogenic in autosomal recessive (AR) diseases. Therefore, in trio ADS were performed to determine in trans or in cis compound heterozygous variants. However, it is not well written in the manuscript.

3. In the introduction section, it is well written about Skeletal dysplasias (SDs, osteochondrodysplasias). However, it is not well written about ATD and the relationship between ATD and SDs are not fully described. Also, it is not well written about rapid WES compound heterozygous mutation, DYNC2H1 gene.   

4. Two variants have registered in dbSNP which is the famous SNP database. It is helpful to add the reference number. The variant (chr11:g.103188645-T>C) is rs1173789302, the variant (chr11:g. 103468635-T>G) is rs1945272232.

5. It was reported that the rs1945272232 variant is associated with ATD (Jeune syndrome).  Have you performed in silico protein functional analysis about the rs1173789302 variant?

6. In the discussion section, the authors presented the various symptoms in ATD (Jeune syndrome), however it has not been compared with this case report. It had better to compare with previous findings. 

7. Several abbreviations are written in the manuscript (For example, AS group in line 52 and JATD in line 204). It is better not to use abbreviations from the beginning.

8. In line 42, “In” should be “in”.

9. In figure 3, the bottom of the letters is cut off.

Author Response

1. The authors used the different disease name, ATD and Jeune syndrome and mentioned ATD, also known as Jeune syndrome. If it is not needed to distinguish, it is better to use either one name.

Answer: Thank you for this suggestion. We used alternative titles of presented entities such as Asphyxiating Thoracic Dystrophy 1 - ATD1 and Jeune syndrome. After your suggestions we made some corrections in the text (in some sentences we transformed the title Jeune syndrome for ATD). But we would like to convince you to keep these alternative titles of this disease.

2. The rapid WES has identified the compound heterozygous variants, however it has not yet been determined whether they are in trans or in cis compound heterozygous variants. It is possible that in cis compound heterozygous variants are not pathogenic in autosomal recessive (AR) diseases. Therefore, in trio ADS were performed to determine in trans or in cis compound heterozygous variants. However, it is not well written in the manuscript.

Answer: as suggested we improved the description of ADS. The corrected part of the Result section is now as follows:To determine if DYNC2H1 compound heterozygous variants are in cis or in trans the tri-based ADS family study was performed. It was found that p.Ile2430Thr variant was inherited from the mother, while p.Leu4239Arg was inherited from the father (Figure 3), which is consistent with in trans variants transmission in the autosomal recessive pattern of inheritance.”

3. In the introduction section, it is well written about Skeletal dysplasias (SDs, osteochondrodysplasias). However, it is not well written about ATD and the relationship between ATD and SDs are not fully described.

Answer:. After your suggestions we made some corrections in the text (Introduction and Discussion). “The asphyxiating thoracic dystrophy known also as Jeune syndrome (ATD; OMIM 208500) belongs to the group of SDs titled as short-rib thoracic dysplasia (https://omim.org/graph/linear/PS208500).“

Also, it is not well written about rapid WES compound heterozygous mutation, DYNC2H1 gene.

Answer: as suggested we described in the Introduction section DYNC2H1 compound heterozygous variants according to Sequence Variant Nomenclature recommendations (HGVS, http://varnomen.hgvs.org/)

The corrected sentence is now as follows:

Herein, we report a case of a foetus showing long-bones shortening and a narrow chest with short ribs, diagnosed prenatally with asphyxiating thoracic dystrophy (ATD; OMIM 208500) caused by compound heterozygous variants (NM_001080463.2:c.[7289T>C];[12716T>G]) in DYNC2H1 gene, identified by prenatally performed rapid-WES analysis.”

4. Two variants have registered in dbSNP which is the famous SNP database. It is helpful to add the reference number. The variant (chr11:g.103188645-T>C) is rs1173789302, the variant (chr11:g. 103468635-T>G) is rs1945272232.

Answer: as suggested we added the RS ID for DYNC2H1 identified variants. And the corrected sentence is now as follows:By applying WES analysis, in the proband two heterozygous missense variants in DYNC2H1 gene were identified: (hg38, chr11:g.103188645-T>C, NM_001080463.2: c.7289T>C; p.Ile2430Thr; rs1221455921) and (hg38, chr11:g. 103468635-T>G, NM_001080463.2: c.12716T>G; p.Leu4239Arg, rs1945272232).”

5. It was reported that the rs1945272232 variant is associated with ATD (Jeune syndrome).  Have you performed in silico protein functional analysis about the rs1173789302 variant?

Answer: according to dbSNP database the rs1173789302 referees to chr5:94537129-T>A (hg38), KIAA0825 gene, intron variant (https://www.ncbi.nlm.nih.gov/snp/rs1173789302#variant_details).

While, DYNC2H1 chr11:g.103188645-T>C variant identified in our study has the following RS ID rs1221455921 (https://www.ncbi.nlm.nih.gov/snp/rs1221455921#variant_details).

We did not perform tailored in silico protein functional analysis for p.Ile2430Thr (rs1221455921). However, since p.Ile2430Thr is located in AAA3 nucleotide binding pocket (amino acids 2251-2505) of DYNC2H1 protein hexomeric ring-like ATP-hydrolysing motor domain (https://www.uniprot.org/uniprot/Q8NCM8) in close proximity of three know Jeune syndrome associated variants, including p.Ser2423Tyr [16], described also as “pathogenic” in ClinVar database (https://www.ncbi.nlm.nih.gov/clinvar) and p.Arg2426Cys [17] and p.Arg2426Leu [16,18]. We consider p.Ile2430Thr variant in combination with Jeune-associated p.Leu4239Arg (rs1945272232) as pathogenic and causative.

6. In the discussion section, the authors presented the various symptoms in ATD (Jeune syndrome), however it has not been compared with this case report. It had better to compare with previous findings.

Answer: After your suggestions we made some corrections in the text (Discussion) “In the presented case, the shortening of the long bones, narrowing of the chest with short ribs, bowing of the femur, excess of soft tissues on the thighs and lower legs, retrognathia, and ulnar deviation of the wrist were noted during ultrasound examination at 14 weeks of gestation.”

7. Several abbreviations are written in the manuscript (For example, AS group in line 52 and JATD in line 204). It is better not to use abbreviations from the beginning.

Answer: Thank you for this suggestions. We corrected those mistakes.

8. In line 42, “In” should be “in”.

Answer: corrected as suggested.

9. In figure 3, the bottom of the letters is cut off.

Answer: we improved Figure 3.